# Deep Ordinal Classification in Forest Areas Using Light Detection and Ranging Point Clouds

**DOI:** 10.3390/s24072168

**Published:** 2024-03-28

**Authors:** Alejandro Morales-Martín, Francisco-Javier Mesas-Carrascosa, Pedro Antonio Gutiérrez, Fernando-Juan Pérez-Porras, Víctor Manuel Vargas, César Hervás-Martínez

**Affiliations:** 1Department of Computer Science and Numerical Analysis, University of Córdoba, Campus de Rabanales, 14071 Córdoba, Spain; pagutierrez@uco.es (P.A.G.); vvargas@uco.es (V.M.V.); chervas@uco.es (C.H.-M.); 2Department of Graphic Engineering and Geomatics, University of Córdoba, Campus de Rabanales, 14071 Córdoba, Spain; ig2mecaf@uco.es (F.-J.M.-C.); o12pepof@uco.es (F.-J.P.-P.)

**Keywords:** LiDAR point cloud, Deep Learning, ordinal classification, soft labeling

## Abstract

Recent advances in Deep Learning and aerial Light Detection And Ranging (LiDAR) have offered the possibility of refining the classification and segmentation of 3D point clouds to contribute to the monitoring of complex environments. In this context, the present study focuses on developing an ordinal classification model in forest areas where LiDAR point clouds can be classified into four distinct ordinal classes: ground, low vegetation, medium vegetation, and high vegetation. To do so, an effective soft labeling technique based on a novel proposed generalized exponential function (CE-GE) is applied to the PointNet network architecture. Statistical analyses based on Kolmogorov–Smirnov and Student’s *t*-test reveal that the CE-GE method achieves the best results for all the evaluation metrics compared to other methodologies. Regarding the confusion matrices of the best alternative conceived and the standard categorical cross-entropy method, the smoothed ordinal classification obtains a more consistent classification compared to the nominal approach. Thus, the proposed methodology significantly improves the point-by-point classification of PointNet, reducing the errors in distinguishing between the middle classes (low vegetation and medium vegetation).

## 1. Introduction

The importance of aerial Light Detection And Ranging (LiDAR), one of the most relevant remote-sensing tools for terrestrial data acquisition, has increased thanks to recent contributions in the field of Artificial Intelligence (AI). Within this area of knowledge, Deep Learning (DL) has played a key role since it has provided the scientific community with further tools to classify and segment 3D point clouds, thus benefiting a wide spectrum of fields, such as Forestry Engineering [1,2], Agricultural Engineering [3,4], and Urban Planning [5,6]. For example, concerning the former, LiDAR technology makes it possible to obtain information at a larger scale. This technology provides information about vegetation structure, density, canopy height model, and canopy percentage cover, which allows a thorough understanding of the forest. To achieve this, LiDAR data need to be previously filtered and classified to analyze forest areas [7].

According to the classification system developed by the American Society for Photogrammetry and Remote Sensing (ASPRS), vegetation can be classified into three classes: low, medium, and high [8]. Low vegetation is that which ranges from ground level up to 0.5 m; medium vegetation is that which ranges from 0.5 m up to 2 m; and high vegetation is that which is more than 2 m [9,10]. Following this differentiation, LiDAR point clouds are used to categorize the points corresponding to the ground in order to generate a Digital Elevation Model (DEM) [11]. Subsequently, thanks to the DEM, the height of the points is normalized, and classified as low vegetation, medium vegetation, or high vegetation. In this context, issues arise when normalized heights are close to 0.5 m or 2 m, as they could be classified as either medium or high vegetation, respectively [12,13]. Once the points are classified, forestry metrics can be obtained, namely percentage cover, tree height, and percentiles [14]. This allows for the calculation of biomass [15], wood estimates [16], or fuel models, all of which help in designing high-precision forest fire prevention and firefighting models [17].

During the last few decades, numerous researchers have focused on classifying and segmenting 3D point clouds using DL approaches. This is because point clouds contain a larger amount of raw information—regarding space, color, etc.—when compared to images. In order to properly classify and segment point clouds, labeling techniques have been considered an essential stage. In the context of labeling point clouds, algorithms can be broadly divided into two groups: those based on methods that operate directly on the 3D point clouds without altering their original nature [18,19], and those based on methods that convert the point clouds to collections of images or voxel grids [20,21]. On the one hand, the first group employs a point-wise discriminative model to assign semantic labels to each element in the point cloud. This model operates on point features and is designed to be simple yet effective. For example, Ref. [22] suggested the use of geometrical and spectral features from the LiDAR point cloud for the semantic labeling task in urban scenarios. On the other hand, the second group employs multiview transformation approaches or volumetric methods to learn local and global features. For instance, Ref. [23] suggested the conversion of the point cloud into regularly distributed 2D images, which allows the classification of a point to be approached as a pixel classification problem. In the case of volumetric methods, Ref. [24] proposed the partition of a given LiDAR point cloud into regular voxel grids, using 3D Convolutional Neural Networks (3D-CNNs) to label each voxel according to the information of its centroid. However, the use of these transformations has been criticized because of the significant computational overhead they introduce, their tendency to increase model complexity, and the potential loss of valuable information.

In this context, to avoid problems with data conversion, many studies have used the PointNet neural network [25,26], which is the pioneering architecture proposed for 3D object recognition in indoor scenarios [27]. This network is specifically designed to process point cloud data, offering a highly adaptable framework with a vast capacity and minimal overhead, enabling efficient operations. For outdoor scenarios, different DL models have been reported, all based on PointNet-like architectures: PointNet++ [28], SE-PointNet++ [29], CropPointNet [30], etc. Meanwhile, other authors have suggested ameliorating the performance of PointNet by exploring the local structure of point clouds [31,32] or incorporating crucial handcrafted features into the deep neural network [33]. Despite all these improvements, the classification and segmentation of large-scale airborne point clouds in complex environments are still challenging [34,35]. For example, in Forestry Engineering, segmenting vegetation represents a challenge due to the intricate interplay between objects and the background, and the need to set thresholds for height differences to normalize and filter LiDAR point clouds. Thus, few studies have assessed the potential of DL-based methods in forest areas, with most of them related to tree species classification [36,37].

In any case, these point cloud labeling algorithms have primarily addressed the resolution of problems where the ordering of the labels has been ignored (nominal classification). Despite the success of the application of DL-based methods in nominal classification tasks in recent years [38,39], ordinal classification has been the focus of researchers in the recognition community. The aim of using ordinal classification is to predict the label of a given pattern, where a natural order among the different possible categories can be assumed. For instance, Ref. [40] aims to predict the human head pose angle (pitch, roll, and yaw) based on ordinal regression techniques with soft labels, operating on 3D point clouds derived from a depth image. In the field of Forestry Engineering, Ref. [41] predicts the number of strata in three datasets based on ordinal regression techniques to support forest management. This task is different from the classification of points as ground, low vegetation, medium vegetation, and high vegetation, which has not yet been explored.

The interest and novelty of the present study lie in developing ordinal classification models for forest areas and integrating them into aerial LiDAR point clouds to refine the per-point classification. The study uses an effective soft labeling technique applied to the PointNet deep network. The main interest of the proposed methodology, compared to the state of the art, is the use of a generalized exponential distribution, where two hyperparameters are introduced (*p* and α) for improving the flexibility of the distribution. By adjusting the value of these parameters, better results can be achieved for the real problem addressed.

The core concept is to enhance the quality of airborne point cloud segmentation by reducing errors in the labeling procedure, such as human operator variability and post-processing steps. The reason for using DL approaches in this research is to explore the potential of this advanced technique to ameliorate the classification of LiDAR data in complex forest environments. The use of DL algorithms can help extract more complex features and patterns from the data, leading to better classification results.

To summarize, this research has focused on the following contributions:The development of an ordinal classification model that utilizes an effective soft labeling technique applied to the loss function using unimodal distributions to ameliorate the classification of LiDAR data.The application of the proposed methodology in forest areas where LiDAR point clouds can be classified into four distinct ordinal labels: ground, low vegetation, medium vegetation, and high vegetation. In the field of Forestry Engineering, it is the first time that this kind of problem is treated as an ordinal regression problem. This contribution is particularly relevant for the forestry industry as it enables more accurate estimation of important forest metrics such as biomass and wood estimates.

This manuscript is organized as follows: the LiDAR point cloud dataset employed, the core concepts such as “ordinal classification” and “soft labeling”, the proposed regularized loss function used for soft labeling (generalized exponential function), and the PointNet network architecture are described in Section 2; the experimental results are shown in Section 3 and discussed in Section 4; and, finally, the conclusions are drawn in Section 5.

## 2. Materials and Methods

### 2.1. LiDAR Data

The LiDAR point cloud generated covered a forest area of 1 ${\mathrm{km}}^{2}$, located in the province of Lugo, Spain ($43^{{}^{\circ}}13^{\prime }50^{\prime \prime }$ N, $7^{{}^{\circ}}55^{\prime }25^{\prime \prime }$ W, WGS-84) (Figure 1). The aerial LiDAR point cloud, which comprises around 46 million points, had a point density of 49 points/$m^{2}$ and a point spacing of 0.143 m with six returns. Moreover, the LASer (LAS) format file of the point cloud contains several point attributes, from which the coordinates (x,y,z), the red, green, and blue color values (RGB), the intensity (*I*), and the return number (Rn) were utilized in this work.

The LiDAR flight was performed on 18 February 2018 using an AS-350 B2 Eurocopter manned helicopter. This platform was equipped with a RIEGL laser scanner, model VQ-480i (Figure 2), which offers an 80 mm footprint. The Inertial Measurement Unit used was iMar Navigation, model iIMU-FSAS-HP-SI-SME1-SP, with an angular measurement range of $\pm 500^{{}^{\circ}}/$s, a drift lower than $0.1^{{}^{\circ}}\times$ ${\mathrm{hour}}^{-1}$, and a resolution of 0.1 arcsec × ${\mathrm{LSB}}^{-1}$ (Least Significant Bit), providing data at a rate of up to 500 Hz. The RGB camera was a Phase One, model IXU-RS 1000, with a resolution of 11,608×8708 pixels and a focal length equal to 50 mm. The GNSS modem utilized was a Javad TR-G3T, incorporating GPS, Galileo, GLONASS, and SBAS frequencies, connecting to up to 32 satellites with an Antcom antenna. The flight altitude of the mission was 300 m above ground level at a speed of 20 m × $s^{-1}$. The system operated at a pulse repetition of 400 kHz and with a scan angle (FOV) of $60^{{}^{\circ}}$, an altitude above ground level equal to 300 m with a reflectivity higher than 20% for a 30% side-lap overlap.

After acquiring the LiDAR data from the experimental site, Python’s pptk package [42] was used to analyze, visualize, and extract features from the 3D point clouds. To ensure the reliability of the labels, conventional and well-established labeling methods were used. In this sense, two filtering methods were applied to label the point clouds as ground truth, using a semi-automatic labeling procedure to evaluate the classification results.

The first one referred to normalized heights of vegetation. Thanks to the Python’s laserchicken package [43], a Digital Elevation Model (DEM) of the study area was generated. In the DEM, each point was assigned a normalized height (*H*) value based on the height of the lowest point in a 1×1 m neighborhood to distinguish vegetation classes. The elevation threshold was established as follows: point clouds from 0 to 0.5 m, from 0.5 to 2 m, and above 2 m were classified as low vegetation, medium vegetation, and high vegetation, respectively.

The second referred to return numbers in order to discriminate ground points from low vegetation points. This filtering was specifically used in areas where only two classes coexist: ground and low vegetation. In these areas, the emitted laser pulse of LiDAR technology first encounters the low vegetation and then the ground class. For this reason, this approach assumes that the first return belongs to the low vegetation class and the other returns belong to the ground class. As previously discussed, the LiDAR point cloud contained six returns. If the return value of a point was higher than or equal to 2, it would mean that the said point could be classified as ground. However, those points with Rn equal to 1 were classified as low vegetation.

All the selected LiDAR point clouds were refined by manually editing them using the interactive segmentation tool of CloudCompare [44]. After that, the LiDAR point clouds were unified and converted into an HDF5 file format [45]. The dataset contained two parts: data and labels. Concerning the former, the attributes of several points from four different categories were recorded. These attributes include x,y,z, RGB values, *I*, *R*, and *H*. In the label section, the labels were categorized into four semantic classes: ground, low vegetation, medium vegetation, and high vegetation (Figure 3).

An amount of 2048 points was randomly sampled from 5738 blocks of size 9×9 m without replacement. After that, the dataset was split into a 60/20/20 percent ratio as training, validation, and test sets, respectively (Table 1). Moreover, all the features were normalized—$x_{n},\;y_{n},\;z_{n},\;R_{n},\;G_{n},\;B_{n},\;I_{n}$ and $\;H_{n}$—except the Rn attribute, which was encoded as a one-hot numeric array—$Rn_{1},\;Rn_{2},\;Rn_{3},\;Rn_{4},\;Rn_{5},\;Rn_{6}$. Lastly, the point clouds in each block constituting the training, validation, and testing sets were converted into an HDF5 file. Each set includes the following: (1) an array of N×2048×14, where *N* is the total number of segmented input blocks, 2048 is the number of points that are randomly sampled per block, and 14 corresponds to the 14-dimensional feature vector (Equation (Equation 1)); and (2) a categorical label encoded as a one-hot numeric array (Equation (Equation 2)).
(1)$$p_{i}=\left(x_{n},\;y_{n},\;z_{n},\;R_{n},\;G_{n},\;B_{n},\;I_{n},\;H_{n},\;Rn_{1},\;Rn_{2},\;Rn_{3},\;Rn_{4},\;Rn_{5},\;Rn_{6}\right).$$
(2)$$C_{q}=\left\{\begin{array}{cc} \left[1,0,0,0\right] & \mathrm{ground}\;\mathrm{category},\\ \left[0,1,0,0\right] & \mathrm{low}\;\mathrm{vegetation}\;\mathrm{category},\\ \left[0,0,1,0\right] & \mathrm{medium}\;\mathrm{vegetation}\;\mathrm{category},\\ \left[0,0,0,1\right] & \mathrm{high}\;\mathrm{vegetation}\;\mathrm{category}, \end{array}\right.$$

### 2.2. Ordinal Classification

The purpose of ordinal classification problems is to predict the real class *y* based on an input *K*-dimensional vector $x(x\in X\subseteq R^{K})$. For this sort of problem, the dependent variable of the classification models is an ordinal scale variable chosen from a set of different categories ($y\in Y=\{C_{1},\;C_{2},\;\ldots ,\;C_{Q}\}$) which have a natural ordering ($C_{1}≺C_{2}≺\ldots ≺C_{Q}$) associated with the real problem [46]. The precedence operator ($C_{i}≺C_{j}$) designates that the category $C_{i}$ is prior to the class $C_{j}$ in the ordinal scale and *Q* is the number of classes defined in the real problem.

Ordinal classification models, which exploit the ordering information in ordinal class attributes, can lead to better segmentation results by reducing misclassification errors in extreme classes, limiting errors to adjacent categories, and maximizing the number of properly classified patterns.

### 2.3. Soft Labeling

Soft labeling is an effective regularization technique that operates on the labels, enhancing the model’s robustness during training. Instead of utilizing hard labels (1 for the target class and 0 for the others), soft labeling assigns some probability to each class [47]. When the labels are encoded as one-hot, the probability distribution of each class is defined as $q\left(i\right)=\delta_{i,c}$, *i* being the predicted class with ranges from 1 to *Q*, *c* the ground truth class, and $\delta_{i,c}$ the Dirac delta, which equals 1 when i=c and 0 otherwise [48].

Based on the above, label smoothing can be incorporated into the standard cross-entropy loss function (Equation (Equation 3)). The cross-entropy loss function is considered one of the most popular methods for training deep neural networks, as it aims to maximize the likelihood of the correct prediction given the ground truth in the training set [49].
(3)$$L=\sum_{i=1}^{Q}q\left(i\right)\left[-\log p\left(y=C_{i}|x\right)\right],$$
where $p(y=C_{i}|x)$ is the probability predicted by the model of *x* belonging to the class *y* for each of the *Q* categories. By applying label smoothing, the q(i) in Equation (Equation 3) can be replaced with a soft version $q^{\prime }\left(i\right)$:(4)$$q^{\prime }\left(i\right)=\left(1-\eta \right)\delta_{i,c}+\eta \left(\frac{1}{Q}\right).$$

A discrete uniform distribution is established associated with the number of classes *Q*, where η is a smoothing parameter defined in the range [0,1] that controls the linear combination.

Within the context of ordinal classification, the misclassification of patterns is a common issue that occurs in adjacent classes in relation to the ground truth on the ordinal scale. To address this challenge and improve the accuracy of loss computation, it is recommended to substitute the use of a uniform distribution with unimodal distributions. The mode of the unimodal distributions should be positioned in the center of the interval for middle classes or at the upper or lower bounds for extreme classes [48]. Additionally, to favor more confident decisions, it is crucial for the probability distribution to exhibit a small variance and for the majority of its probability mass to be concentrated within the interval corresponding to the ground truth class [48].

Various low-variance distribution functions considered in previous studies [48,50,51,52] have ameliorated the performance of ordinal classifiers by applying a soft labeling technique regarding the standard one-hot encoding. Some of the proposed methods require experimental adjustment of different parameters. Ref. [50] proposed using a binomial distribution, which takes into account the number of classes and the probability. The mean and variance of the binomial distribution are given by different expressions, allowing for enhanced flexibility to place the mean in the center of the class interval while keeping the variance small. According to [51], the Poisson distribution represents an alternative for modeling the probabilities. However, since the mean and variance of the Poisson distribution are both determined by the parameter λ, it is not possible to center the mean of the distribution in the class interval while achieving a small variance, leading to poor performance. In the work of [52], an exponential function followed by a Softmax normalization is explained. Besides its simplicity, this distribution flexibly adjusts the shape of the target label distribution in order to smooth the label with the unimodal distribution. Hence, it is feasible to manipulate the mean and variance of the exponential distribution to foster label predictions that are closely distributed to the ground truth class. Lastly, Ref. [48] proposed sampling from a beta distribution that is defined within the range of 0 to 1; thus, no normalization is mandatory and no high variance is achieved. However, an improvement of this distribution could be considered when the number of classes is low.

### 2.4. Generalized Exponential Function

All the abovementioned distributions, especially when the number of classes is low, are characterized by high variance or lack the necessary flexibility to center the mode of each distribution within the interval corresponding to its class. For this reason, a generalized exponential function based on [52] is proposed:(5)$$f\left(q,p,\alpha \right)=e^{{-\alpha |q-y|}^{p}},$$
where *q* and *y* designate the predicted and real classes, respectively, and 1≤p≤2 and 0<α≤2 are two hyperparameters that need to be adjusted experimentally. After that, a Softmax normalization procedure is employed to calculate the corresponding probabilities. Figure 4 presents the class distributions for the proposed generalized exponential function, where the color indicates the ground truth class of a given pattern, the *x*-axis specifies the class under examination, and the *y*-axis indicates the applied soft label. For each class, the distributions for p∈{1.0,1.5,2.0} and α∈{0.5,1.0,1.5,2.0} are represented.

These probabilities replace the uniform distribution and are subsequently used to generate the corresponding $q^{\prime }\left(i\right)$ (Equation (Equation 4)). In this way, the standard definition of the loss function is formulated as:(6)$$L=\sum_{i=1}^{Q}q^{\prime }\left(i\right)\left[-\log p\left(y=C_{i}|x\right)\right].$$

Given that $q^{\prime }\left(i\right)$ presents a continuous decrease with respect to the farther distance from the ground truth class, it is considered as a weight of $-\log p(y=C_{i}|x)$.

In the present study, the uniform distribution is replaced with other low-variance distributions that provide more flexibility when centering each of the four categories in the problem. In these low-variance distributions, f(x,θ) is the probability value sampled from a binomial, Poisson, beta, and generalized exponential plus a Softmax function:(7)$$q^{\prime }\left(i\right)=\left(1-\eta \right)\delta_{i,c}+\eta f\left(x,\theta \right).$$

To analyze the behavior of the proposed distribution, an ordinal classification model with four classes (Q=4) is considered. To find the hyperparameters (p,α) from the generalized exponential function, one value is decided while fixing the other one, reducing the computational cost of simultaneously finding both values. The value for the hyperparameter *p* is obtained by fixing the hyperparameter α (Equation (Equation 8)). After that, to obtain the value of the hyperparameter α, the hyperparameter *p* is fixed (Equation (Equation 9)).
(8)$$f\left(q,\;p,\;\alpha =1\right)=\;e^{{-|q-y|}^{p}},\;q=1,\;\ldots ,\;Q.$$
(9)$$f\left(q,\;p=1,\;\alpha \right)=\;e^{-\alpha |q-y|},\;q=1,\;\ldots ,\;Q.$$

### 2.5. PointNet Network Architecture

A Convolutional Neural Network (CNN), namely PointNet [27], is used for the classification of LiDAR point clouds, as the main interest of this study is to offer a point-by-point classification. Nevertheless, some modifications in the inputs and outputs are included for the proper implementation of the original network [27,31].

Firstly, all the point features employed in the present study—the coordinates x,y, and *z*; the spectral values red, green, and blue (RGB); the intensities (*I*); and the height of each point within the 1×1 m block (*H*)—are normalized—$x_{n},\;y_{n},\;z_{n},\;R_{n},\;G_{n},\;B_{n},\;I_{n}$ and $\;H_{n}$—instead of only adding only the normalized coordinates of the point [31].

Secondly, considering that the return number attribute (Rn) is a discrete variable defined in the range [1,6] (the LiDAR system provides six returns), it is encoded as a one-hot integer array—$Rn_{1},\;Rn_{2},\;Rn_{3},\;Rn_{4},\;Rn_{5},\;Rn_{6}$. Thus, this encoding allows for evaluating the impact of Rn in the model, as the other parameters are measured on a continuous scale.

Thirdly, prior to data training and following the design of the PointNet model, the point cloud is partitioned into non-overlapping blocks of size 9×9 m. The number of points to be randomly sample per block is set to 2048 in order to define consistent data batches. According to [31], if the number of points within the block exceeds the maximum allowed, a random selection process is employed to determine which points to include. Hence, the network is fed by a B×N×2048×14 array, where *B* denotes the batch size, *N* denotes the total number of segmented input blocks, 2048 denotes the number of points randomly sampled per block, and 14 represents the 14-dimensional feature vector ($x_{n},\;y_{n},\;z_{n},\;R_{n},\;G_{n},\;B_{n},\;I_{n},\;H_{n},Rn_{1},\;Rn_{2},\;Rn_{3},\;Rn_{4},\;Rn_{5},\;Rn_{6}$).

Fourthly, as the described model is intended to segment 3D point clouds, a vector output is obtained for each point that contains the probabilities of each class for the given point.

Lastly, a Softmax function is used in the output layer to attain the probability of point cloud prediction in each class within the range [0, 1]: ground, low vegetation, medium vegetation, and high vegetation (Figure 5). In addition, a dropout rate of 0.3 is applied for the last two dense layers with ReLU activation [27].

### 2.6. Experiment Settings

The loss function was weighted following the methodology described in [48] to alleviate the imbalance of different categories: the minority classes had a higher weight than the main classes.

Five different loss functions used for the optimization algorithm and derived from the standard cross-entropy loss were established: categorical cross-entropy (CCE), cross-entropy loss with binomial regularization (CE-B) [52], cross-entropy loss with Poisson regularization (CE-P) [52], cross-entropy loss with β regularization (CE-β) [48], and the proposed cross-entropy loss with generalized exponential regularization (CE-GE).

A confusion matrix Q×Q was used to analyze the potential of the classifiers (Table 2). In the context of an ordinal classification problem involving *Q* categories and *n* patterns, $n_{qk}$ denotes the frequency with which a classifier assigns patterns belonging to class *q* to class *k*, $n_{q•}$ denotes the total number of patterns in class *q*, and $n_{•k}$ denotes the total number of patterns predicted to be in class *k* [46].

Different metrics chosen for evaluation were applied to quantitatively compare and analyze the classification performance. These metrics included the Quadratic Weighted Kappa (QWK), the Minimum Sensitivity (MS), the Mean Absolute Error (MAE), the Correct Classification Rate (CCR), the 1-off accuracy (1-off) and the Mean Intersection-Over-Union (mIoU) [46,48]. Thanks to Python’s dlordinal package [53], the ordinal metrics MS and 1-off were computed.
(10)$$\mathrm{QWK}=\frac{\rho_{o(w)}-\rho_{e(w)}}{1-\rho_{e(w)}},$$
(11)$$\rho_{o(w)}=\frac{1}{n}\sum_{q=1}^{Q}\sum_{k=1}^{Q}w_{qk}n_{qk},$$
(12)$$\rho_{e(w)}=\frac{1}{n^{2}}\sum_{q=1}^{Q}\sum_{k=1}^{Q}w_{qk}n_{q•}n_{•k},$$
(13)$$\mathrm{MS}=\min\left\{S_{q}=\frac{n_{qq}}{n_{q•}};q=1,\cdots ,Q\right\},$$
(14)$$\mathrm{MAE}=\frac{1}{n}\sum_{q,k=1}^{Q}\left|q-k\right|n_{qk}=\frac{1}{n}\sum_{i=1}^{n}e\left(x_{i}\right),$$
(15)$$\mathrm{CCR}=\frac{1}{n}\sum_{q=1}^{Q}n_{qq},$$
(16)$$1-\mathrm{off}=\frac{1}{n}\sum_{q=1}^{Q}\sum_{k=max(0,q-1)}^{min(Q,q+1)}n_{qk},$$
(17)$$mIoU=\frac{1}{n}\sum_{q=1}^{Q}\frac{S_{qq}}{\sum_{k=1}^{Q}S_{qk}+\sum_{k=1}^{Q}S_{kq}-S_{qq}},$$
where $w_{qk}=\left|q-k\right|$, *q* and *k* being the true and the predicted category, respectively; $S_{q}$ denotes the sensitivity of the *q*-th class; $e\left(x_{i}\right)=\left|O\left(y_{i}\right)-O\left(y_{i}^{*}\right)\right|$ denotes the distance between the true ($y_{i}$) and the predicted ($y_{i}^{*}$) ranks; $O(C_{q})=q$ denotes the position of each category in the ordinal scale; $S_{qq}$, $S_{qk}$, and $S_{kq}$ denote the number of true positives, false positives, and false negatives, respectively.

The proposed methodology was implemented based on the TensorFlow Keras frameworks using the novel ordinal dataset. The experiments followed a hold-out scheme which was executed 20 times with 20 distinct seeds to ensure a comprehensive evaluation. This rigorous approach enabled the attainment of 20 independent executions. The training process was run for 150 epochs with a batch size of 64 and a momentum of 0.9 [54]. The number of training epochs and the batch size were increased with respect to the previous study [54], from 30 to 150 and from 32 to 64, respectively, due to the amount of points per-block sampled. The model was optimized using the Adam method [55], with an initial learning rate of 0.001. In addition, decay on the learning rate was also employed when the validation loss had not decreased for 10 epochs, multiplying it by a factor of 0.85 until it reached $10^{-7}$.

## 3. Results

### 3.1. Calculating p and α from Generalized Exponential Function

A preliminary experiment was conducted to obtain the hyperparameters *p* and α from the generalized exponential function (Appendix A). In Table A1, the results obtained by defining *p* as p∈{1.0,1.1,1.2,1.3,1.4,1.5,1.6,1.7,1.8,1.9,2.0} showed that the optimum value of *p* was 1, based on the six metrics previously described. After that, the value of the hyperparameter α was also selected by fixing p=1. Likewise, as shown in Table A2, the results obtained by defining α as α∈{0.25,0.50,0.75,1.00,1.25,1.50,1.75,2.00} showed that the optimum value was 1.

Considering these results, the performance of each method is evaluated in Table 3. All the values introduced include the average and standard deviation of the 20 executions. The best result for each metric is highlighted in bold font, while the second best is in italics. Hence, the CE-GE method exhibited the best mean results compared to other methodologies.

### 3.2. Statistical Analysis

To ensure the reliability of these results, statistical analyses were conducted, in which each of the metrics presented in Section 2.6 was studied independently. The Kolmogorov–Smirnov test [56] for the QWK, MS, MAE, CCR, 1-off, and mIoU was performed using the results obtained from the 20 executions. The *p*-values achieved were higher than 0.050, indicating that the null hypothesis of normality for the distribution of the values of these metrics was accepted. Consequently, each pair of algorithms, CE-GE with the other methods, was compared for each metric by employing Student’s *t*-test for paired data [56]. A significance level of α=0.100 was established and its corresponding adjustment based on the number of comparisons was added. Since four algorithms were compared, the corrected significance level was calculated ($\alpha =\frac{0.100}{4}=0.025$).

Table 4 shows the mean and the standard deviation (SD) of each method’s executions on the test set, together with the *p*-values of the test. In general terms, statistically significant differences were observed in the results of the paired *t*-test. For all of the metrics, the CE-GE method exhibited the best mean results. Taking into account the QWK and 1-off accuracy metrics, the CE-GE achieved better mean results than CCE, CE-B, and CE-P for α=0.050 and CE-β for α=0.100. On the other hand, for the MS and mIoU metrics, the CE-GE reached better mean results than CE-B and CE-P for α=0.050 and CE-β for α=0.100. Finally, based on the MAE and CCR metrics, the CE-GE got better mean results than the other methods for α=0.050.

### 3.3. LiDAR Classification

The performance of PointNet for each category of points is evaluated by analyzing the confusion matrices of two methods: the proposed best alternative (CE-GE + Softmax) and the standard method (CCE + Softmax). Normalized confusion matrices from the smoothed ordinal classification and the nominal classification are shown in Figure 6. Additionally, Table 5 and Table 6 present an in-depth analysis of the confusion matrices.

Using nominal classification (Figure 6b and Table 5), the ground class was the only one correctly labeled when compared to the confusion matrix from the smoothed ordinal classification. In this sense, in nominal classification, distinguishing between ground and vegetation classes is easier than in ordinal classification because the former is treated as a binary classification problem (e.g., ground vs. vegetation). On the other hand, smoothed ordinal classification uses the ordering information between the four classes, reinforcing the decision boundaries among low, medium, and high vegetation classes. Therefore, in the case of the smoothed ordinal classification (Figure 6a and Table 6), points classified as low vegetation, medium vegetation, and high vegetation were correctly labeled for the majority of the test set.

As an example, Figure 7 shows a partial view of the segmentation results of the PointNet model in the study area. As can be observed, the best alternative proposed, CE-GE + Softmax, which incorporates an ordinal structure and effective soft labeling simultaneously, performed better than the standard method CCE + Softmax. As previously mentioned, most of the differences obtained by the nominal and the smoothed classification results are related to the misclassification of the middle classes. In Figure 7, the main differences between the ground truth, the nominal method (CCE), and the ordinal methodologies (CE-β and CE-GE) were depicted. The proposed method CE-GE + Softmax was closer to the ground truth than both the nominal point cloud-based method and the beta ordinal method as it generated the correct label predictions for most point clouds.

## 4. Discussion

The classification of LiDAR point clouds has gained popularity due to the high demand for accurate classifications. To contribute to this field, previous research has increasingly utilized DL methods, training models with different point attributes to simulate operator labeling [57]. Nevertheless, most studies have focused on point attributes rather than the nature of the labels themselves [38,39]. This could be because most studies have focused on urban scenarios where large-scale point clouds often contain distinct classes such as ground, vegetation, building, or water, which can easily be distinguished based on LiDAR point characteristics [58]. In these scenarios, previous studies have achieved an overall accuracy of 87% in classifying the following classes: ground, vegetation, and building [31,32].

In this study, the classification of LiDAR point clouds in forest areas was addressed. In these areas where vegetation is the dominant class, it becomes crucial to accurately distinguish between different vegetation strata [6]. Points belonging to the middle classes are difficult to classify because their normalized height presents similar geographical distribution and topological features compared to other categories [6]. In this sense, the low vegetation class can easily be confused with ground or medium vegetation, and the medium vegetation category can be difficult to differentiate from low vegetation or high vegetation. In our proposal, to determine the most suitable approach for accurately classifying these categories, five different methodologies (CCE, CE-B, CE-P, CE-β, and CE-GE) were compared [48,50,51,52].

Based on the results obtained, the CE-GE methodology was found to be the most effective approach for tackling this classification problem. By assigning a probability to the classes (soft labeling) and considering the order between the categories, the issue of misclassification regarding the middle classes (low vegetation and medium vegetation) was mitigated. Utilizing the CE-GE methodology, the study achieved an overall accuracy of 95% in classifying the following classes: ground, low vegetation, medium vegetation, and high vegetation. Nevertheless, it is important that the data must have a clear ordering between the different classes, which is the main challenge that may arise when applying the ordinal model.

For those cases where an ordinal structure is not found in the labels, future projects could incorporate this methodology as a pre-classification technique for the following four classes: ground, low vegetation, medium vegetation, and high vegetation. For upcoming projects, the proposed methodology could explore agricultural areas and urban green spaces, in addition to forest scenarios.

## 5. Conclusions

The present study focused on optimizing the existing DL network PointNet for classification in complex environments. It analyzed the effects of adding an ordinal classification and an effective soft labeling technique to point-by-point classification. Specifically, for four categorical classes, the study proposed the application of a CE-GE function, which helps achieve more accurate labels. In addition, a comparison was performed with different loss functions (CCE, CE-B, CE-P, and CE-β) to demonstrate the benefits of the smoothed ordinal classification for the PointNet model.

Given the results obtained, the following conclusions should be highlighted:Significant differences in the paired *t*-test were observed, in which the CE-GE method reached the best mean results for all the metrics (QWK, MS, MAE, CCR, and 1-off) when compared to the other methods.Regarding the confusion matrices of the best alternative conceived (CE-GE + Softmax) and the standard method (CCE + Softmax), the smoothed ordinal classification achieved notably better results than the nominal one. Consequently, our methodology reduce the errors in distinguishing between the middle classes (low vegetation and medium vegetation).

Overall, the proposed methodology has significantly improved the point-by-point classification of PointNet. The application of the suggested methodology in forest areas was possible, contributing to obtaining forestry metrics up ahead, which allows thorough understanding of the forest, as it permits the calculation of biomass, wood estimates, and fuel models for high-precision forest fire prevention and firefighting. Therefore, as long as LiDAR point clouds have an underlying ordinal structure, they could be used in a diverse range of forest ecology and management applications, from monitoring land-cover changes in complex environments to identifying tree species in future scenarios.

## Figures and Tables

**Figure 1 sensors-24-02168-f001:**
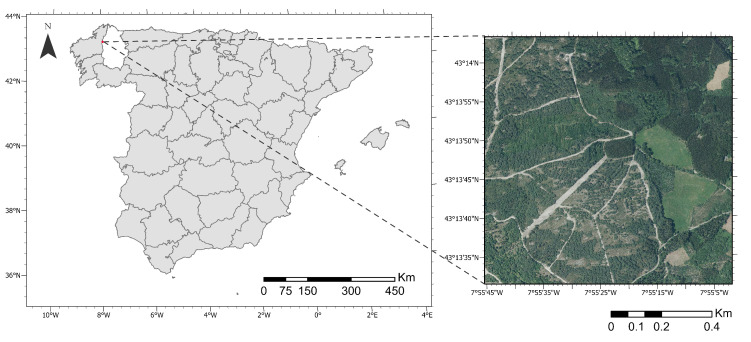
Location of the LiDAR data. The coordinates are georeferenced in the ETRS89/UTM Zone 29N coordinate system.

**Figure 2 sensors-24-02168-f002:**
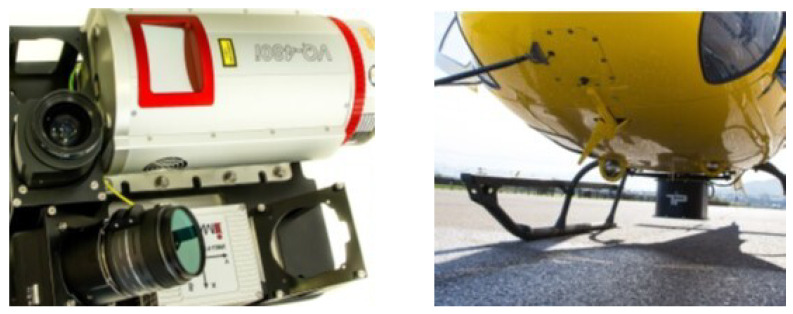
On the left, the laser scanner, two Phase One cameras (zenith and oblique), and the IMU. On the right, the system installed before the flight.

**Figure 3 sensors-24-02168-f003:**
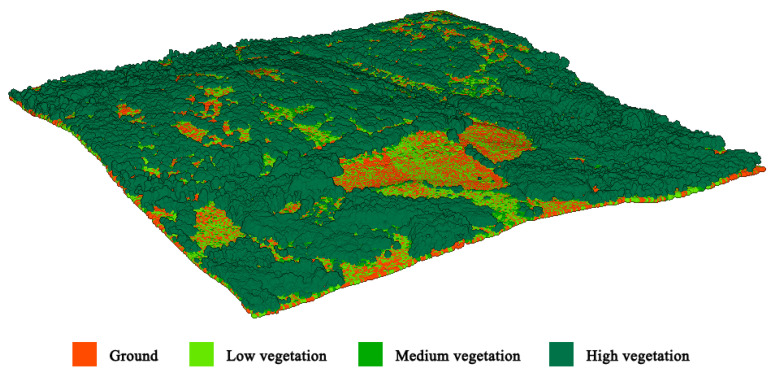
Labels of the LiDAR point cloud. The legend represents the color corresponding to each category.

**Figure 4 sensors-24-02168-f004:**
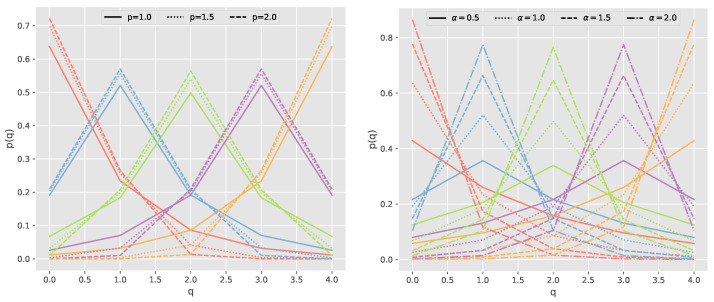
The generalized exponential distribution was used to address a classification problem with five distinct classes. In the resulting visualization, the *x*-axis designates the evaluated class, while the *y*-axis displays the corresponding smoothed label value. The color coding corresponds to the ground truth class, with red representing class 0, blue representing class 1, green representing class 2, purple representing class 3, and orange representing class 4. Each line in the visualization represents the probability distribution for a specific true label. The line type indicates the different values that the hyperparameter *p* and the hyperparameter α can take.

**Figure 5 sensors-24-02168-f005:**
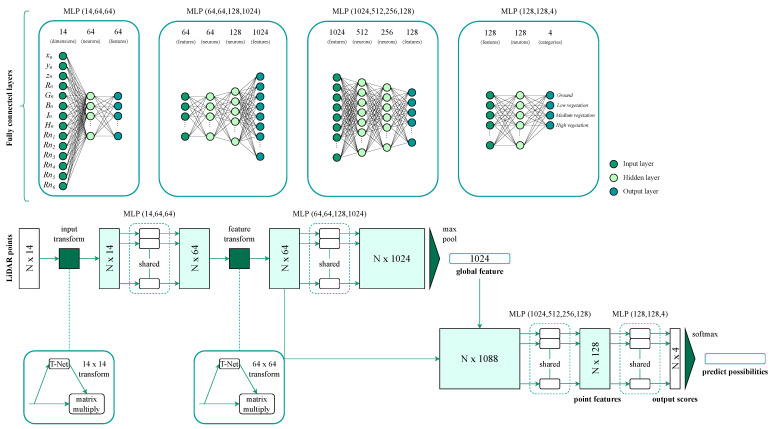
The architecture of the refined DL network PointNet. In the resulting visualization, there are four boxes in the upper part displaying the MLP layers and two boxes in the lower part displaying the T-Net layers. The CNN processes the 14-dimensional features that contain *N* grids and 2048 points within the block. The T-net layers are responsible for transforming the input and point features, while the MLPs and the max-pooling layers are responsible for aggregating and extracting the point features. The individual point-level and global features are concatenated to acquire the point features. The output includes the predicted probabilities of belonging to each of the four classes.

**Figure 6 sensors-24-02168-f006:**
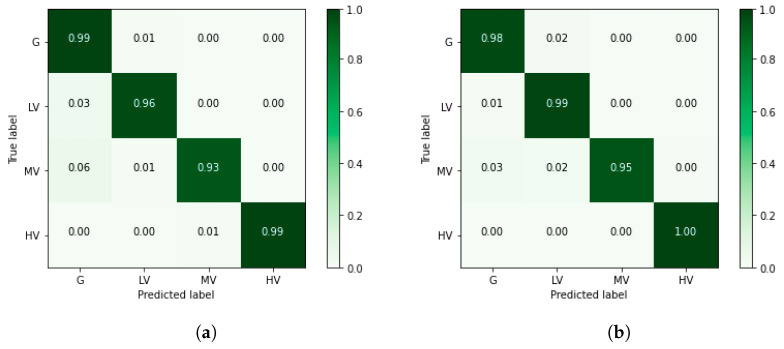
Normalized confusion matrices from (**a**) the nominal classification and (**b**) the smoothed ordinal classification. Each number represents the ratio of patterns to the total number of patterns in the class. G: ground; LV: low vegetation; MV: medium vegetation; and HV: high vegetation.

**Figure 7 sensors-24-02168-f007:**
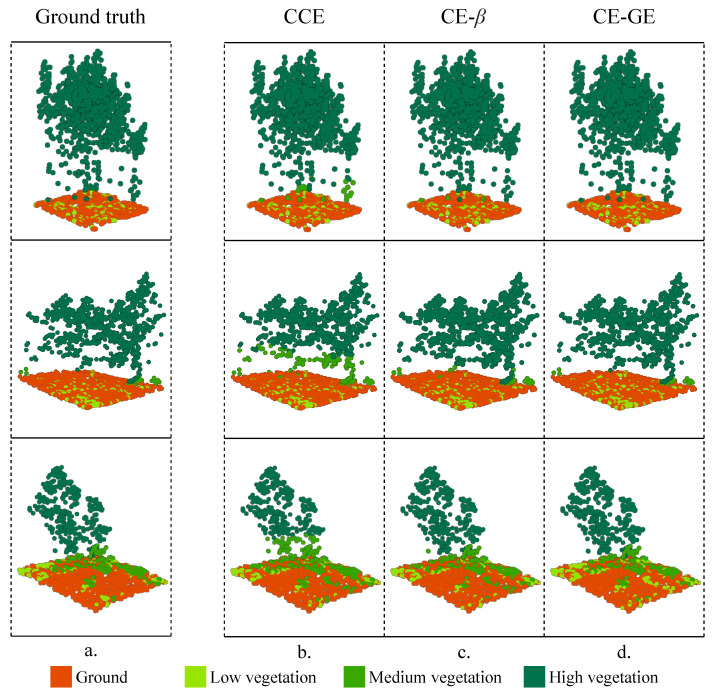
Segmentation result: (**a**) ground truth, (**b**) nominal methodology (CCE), (**c**) ordinal methodology based on beta distribution (CE-β), and (**d**) proposed ordinal methodology based on generalized exponential distribution (CE-GE).

**Table 1 sensors-24-02168-t001:** The number of points in each category.

Category	Training	Validation	Test
Ground	2,578,600	863,096	835,056
Low vegetation	1,000,907	324,547	354,049
Medium vegetation	1,204,729	412,513	400,521
High vegetation	2,264,980	750,948	761,478
	7,049,216	2,351,104	2,351,104

**Table 2 sensors-24-02168-t002:** Confusion matrix.

		Predicted Class					
		**1**	⋯	k	⋯	Q	
	1	$n_{11}$	⋯	$n_{1k}$	⋯	$n_{1Q}$	$n_{1•}$
		⋮		⋮		⋮	⋮
**True class**	*q*	$n_{q1}$	⋯	$n_{qk}$	⋯	$n_{qQ}$	$n_{q•}$
		⋮		⋮		⋮	⋮
	*Q*	$n_{Q1}$	⋯	$n_{Qk}$	⋯	$n_{QQ}$	$n_{Q•}$
		$n_{•1}$	⋯	$n_{•k}$	⋯	$n_{•Q}$	*n*

Note. Table adapted from [46].

**Table 3 sensors-24-02168-t003:** Comparison of methodologies for each metric.

Method ^1^	QWK	MS	MAE	CCR	1-off	mIoU
CCE	*0.9535_0.0258_*	*0.8077_0.1281_*	*0.0920_0.0495_*	*0.9378_0.0310_*	*0.9703_0.0187_*	*0.8757_0.0393_*
CE-B	$0.9460_{0.0277}$	$0.7687_{0.1301}$	$0.1095_{0.0585}$	$0.9244_{0.0402}$	$0.9665_{0.0189}$	$0.8419_{0.0800}$
CE-P	$0.9395_{0.0485}$	$0.7451_{0.2223}$	$0.1198_{0.0951}$	$0.9209_{0.0562}$	$0.9609_{0.0361}$	$0.8381_{0.1054}$
CE-β	$0.9504_{0.0335}$	$0.7746_{0.1702}$	$0.0999_{0.0674}$	$0.9320_{0.0422}$	$0.9682_{0.0253}$	$0.8583_{0.0817}$
CE-GE	$0.9667_{0.0119}$	$0.8515_{0.0600}$	$0.0667_{0.0225}$	$0.9539_{0.0147}$	$0.9794_{0.0084}$	$0.8885_{0.0303}$

^1^ CCE: categorical cross-entropy; CE-B: cross-entropy loss with binomial regularization; CE-P: cross-entropy loss with Poisson regularization; CE-*β*: cross-entropy loss with *β* regularization; and CE-GE: cross-entropy loss with generalized exponential regularization.

**Table 4 sensors-24-02168-t004:** Paired sample *t*-test results for the QWK, MS, MAE, CCR, 1-off, and mIoU metrics.

Metric	Paired Sample ^1^	Mean	SD	*p*-Value
QWK	CE-GE vs CCE	0.014	0.026	0.024
	CE-GE vs. CE-β	0.017	0.041	0.081
	CE-GE vs. CE-B	0.021	0.027	0.003
	CE-GE vs. CE-P	0.028	0.052	0.023
MS	CE-GE vs CCE	0.044	0.142	0.182
	CE-GE vs. CE-β	0.077	0.172	0.061
	CE-GE vs. CE-B	0.083	0.130	0.010
	CE-GE vs. CE-P	0.106	0.216	0.040
MAE	CE-GE vs CCE	−0.025	0.050	0.034
	CE-GE vs. CE-β	−0.033	0.071	0.050
	CE-GE vs. CE-B	−0.043	0.055	0.002
	CE-GE vs. CE-P	−0.053	0.094	0.021
CCR	CE-GE vs CCE	0.016	0.030	0.024
	CE-GE vs. CE-β	0.022	0.046	0.044
	CE-GE vs. CE-B	0.030	0.039	0.003
	CE-GE vs. CE-P	0.033	0.056	0.017
1-off	CE-GE vs CCE	0.010	0.019	0.033
	CE-GE vs. CE-β	0.012	0.030	0.089
	CE-GE vs. CE-B	0.014	0.018	0.003
	CE-GE vs. CE-P	0.019	0.038	0.035
mIoU	CE-GE vs CCE	0.013	0.049	0.174
	CE-GE vs. CE-β	0.030	0.083	0.082
	CE-GE vs. CE-B	0.047	0.082	0.012
	CE-GE vs. CE-P	0.050	0.111	0.022

^1^ CCE: categorical cross-entropy; CE-B: cross-entropy loss with binomial regularization; CE-P: cross-entropy loss with Poisson regularization; CE-*β*: cross-entropy loss with *β* regularization; and CE-GE: cross-entropy loss with generalized exponential regularization.

**Table 5 sensors-24-02168-t005:** Confusion matrix from the nominal classification results.

	Ground	Low Vegetation	Medium Vegetation	High Vegetation
Ground	**827,294**	7667	95	0
Low vegetation	12,368	**341,629**	52	0
Medium vegetation	23,385	4259	**372,858**	19
High vegetation	0	0	8083	**753,395**

**Table 6 sensors-24-02168-t006:** Confusion matrix from the smoothed ordinal classification results.

	Ground	Low Vegetation	Medium Vegetation	High Vegetation
Ground	**815,576**	17,125	2355	0
Low vegetation	3973	**350,035**	41	0
Medium vegetation	12,363	7723	**378,657**	1778
High vegetation	0	0	1258	**760,220**

## Data Availability

The datasets presented in this article are not readily available because the data are part of an ongoing study.

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
