# Peer review of "Deep Ordinal Classification in Forest Areas Using Light Detection and Ranging Point Clouds"

_sensors, 2024, doi:10.3390/s24072168_

Round 1
Reviewer 1 Report
Comments and Suggestions for Authors
This paper propose an effective soft labeling technique that applied to the PointNet network for ordinal classification in complex forest areas. Howerver there are some problems to be solved:
1. The proposed method is simple. And the nolvety should be further improved.
2. The experimental results are not convincing. For example, the results in Fig.7 are poor. More segmentation results should be provided.
3. This paper focuses on forest areas, but there are not enough data to prove the effectiveness.
4. The experimental comparison is not sufficient, and it is recommended to add more methods for comparative experiments,especially the visualization result.
Author Response
Response to Reviewers’ comments to manuscript sensors-2905206
The authors gratefully acknowledge the Associate Editor and the Anonymous Reviewers for their detailed and highly constructive criticisms, which greatly helped us to improve the quality and presentation of our manuscript. In the following, we provide detailed, item-by-item, point by point responses to all the very interesting issues raised by the Anonymous Reviewers. We would also like to emphasize that, in order to simplify the review of our manuscript, we have highlighted the main modifications introduced in the reviewed manuscript in blue color to help the Associate Editor and the Anonymous Reviewers find the changes made with regards to the previous version. We are indebted to them for their careful assessment and outstanding suggestions for improving our manuscript, which have been really helpful in order to enhance its presentation and technical quality.
Responses to the comments by Reviewer 1
This paper propose an effective soft labeling technique that applied to the PointNet network for ordinal classification in complex forest areas. However there are some problems to be solved.
The authors would like to express their deep gratitude to the Anonymous Reviewer for their valuable time and effort spent on enhancing this manuscript. We have addressed each of their comments and suggestions in the following item-by-item response.
The proposed method is simple. And the novelty should be further improved.
Following the Reviewer’s suggestion, the authors have refined the introduction section of the manuscript in the reviewed version to highlight the novelty of this research.
"The interest and novelty of the present study lies in developing ordinal classification models for forest areas and integrating them into aerial LiDAR point clouds to refine the per-point classification. The study uses an effective soft labeling technique applied to the PointNet deep network. The main interest of the proposed methodology, compared to the state-of-the-art, is the use of a Generalized Exponential distribution, where two hyperparameters are introduced (p and α) for improving the flexibility of the distribution. By adjusting the value of these parameters, better results can be achieved for the real problem addressed." (line 90 to 97).
The experimental results are not convincing. For example, the results in Fig.7 are poor. More segmentation results should be provided.
The Reviewer’s suggestion has been considered and implemented in Figure 7 of the reviewed manuscript (page 14), where the authors have included more methods and examples in the segmentation results.
This paper focuses on forest areas, but there are not enough data to prove the effectiveness.
The Reviewer is correct, this methodology has focused on forest areas, but not in agricultural areas or urban green spaces. In the case of forest areas, given their spatial heterogeneity, adding more data involves training new models, because it is necessary to adapt them to the new environmental conditions. For this reason, the methodology proposed has been validated in a large number of samples with a high density.
For clarity, the authors have incorporated the suggested content into the reviewed manuscript.
"For those cases where an ordinal structure is not found in the labels, future projects could incorporate this methodology as a pre-classification technique for the following four classes: ground, low vegetation, medium vegetation, and high vegetation. For upcoming projects, the proposed methodology could explore agricultural areas and urban green spaces, in addition to forest scenarios." (line 418 to 422).
The experimental comparison is not sufficient, and it is recommended to add more methods for comparative experiments, especially the visualization result.
Five methodologies (the standard nominal cross entropy error and the four versions of unimodal regularized networks, see Table 3) were compared in this research. However, it is certainly true that visualisation results were not discussed enough given that we only compared two methods. Consequently, the Reviewer’s suggestion has been considered and implemented in the reviewed version of Figure 7 of the manuscript, including now the second best performing method (the beta distribution).
"In Figure 7, the main differences between the ground truth, the nominal method (CCE), and the ordinal methodologies (CE-β and CE-GE) were depicted. The proposed method CE-GE + Softmax was closer to the ground truth than both the nominal point cloud-based method and the beta ordinal method as it generated the correct label predictions for most point clouds." (line 384 to 388).

Reviewer 2 Report
Comments and Suggestions for Authors
The authors developed an ordinal classification model in forest areas, and applied a soft labeling technique to point-by-point classification. The methodology improved the point-by-point classification of PointNet, reducing the errors in distinguishing between the middle classes. The experiment is very interesting and provides an effective means of target differentiation in complex environments. I recommend this article for publication in Sensors with some minor revisions.
1. Line 360 to 364, “Using nominal classifcation (Figure 6b and Table 5), the ground class was the only one correctly labeled when compared to the confusion matrix from the smoothed ordinal classification.” Why using the smoothed ordinal classification significantly improve the test results. The authors should add some explanation.
2. Line 383 “In these scenarios, previous studies have achieved an overall accuracy of 87% in classifying the following classes: ground, vegetation, and building” How about the accuracy of the method now proposed?
Author Response
Response to Reviewers’ comments to manuscript sensors-2905206
The authors gratefully acknowledge the Associate Editor and the Anonymous Reviewers for their detailed and highly constructive criticisms, which greatly helped us to improve the quality and presentation of our manuscript. In the following, we provide detailed, item-by-item, point by point responses to all the very interesting issues raised by the Anonymous Reviewers. We would also like to emphasize that, in order to simplify the review of our manuscript, we have highlighted the main modifications introduced in the reviewed manuscript in blue color to help the Associate Editor and the Anonymous Reviewers find the changes made with regards to the previous version. We are indebted to them for their careful assessment and outstanding suggestions for improving our manuscript, which have been really helpful in order to enhance its presentation and technical quality.
Responses to the comments by Reviewer 2
The authors developed an ordinal classification model in forest areas, and applied a soft labeling technique to point-by-point classification. The methodology improved the point-by-point classification of PointNet, reducing the errors in distinguishing between the middle classes. The experiment is very interesting and provides an effective means of target differentiation in complex environments. I recommend this article for publication in Sensors with some minor revisions.
The authors would like to take this opportunity to gratefully thank the Anonymous Reviewer for the suggestions and comments for improvement that they provided. Below we provide an item-by-item response to each of these comments and suggestions.
Line 360 to 364, “Using nominal classification (Figure 6b and Table 5), the ground class was the only one correctly labeled when compared to the confusion matrix from the smoothed ordinal classification.” Why using the smoothed ordinal classification significantly improve the test results. The authors should add some explanation.
The authors gratefully acknowledge for pointing this out. In nominal classification, distinguishing between ground and vegetation classes is easier than in ordinal classification because the former is treated as a binary classification problem (e.g., ground vs. vegetation). By contrast, smoothed ordinal classification utilizes the ordering information between the classes, reinforcing the decision boundary between low, medium, and high vegetation.
Following the Reviewer’s suggestion, the authors have decided to include the suggested content into the reviewed manuscript.
"In this sense, in nominal classification, distinguishing between ground and vegetation classes is easier than in ordinal classification because the former is treated as a binary classification problem (e.g., ground vs. vegetation). On the other hand, smoothed ordinal classification uses the ordering information between the four classes, reinforcing the decision boundary among low, medium, and high vegetation classes." (line 371 to 375).
Line 383 “In these scenarios, previous studies have achieved an overall accuracy of 87% in classifying the following classes: ground, vegetation, and building” How about the accuracy of the method now proposed?
As suggested by the Reviewer, the authors have incorporated the overall accuracy in the reviewed manuscript.
"Utilizing the CE-GE methodology, the study achieved an overall accuracy of 95% in classifying the following classes: ground, low vegetation, medium vegetation, and high vegetation." (line 413 to 415).

Reviewer 3 Report
Comments and Suggestions for Authors
In this paper, the authors present "Deep Ordinal Classification in Forest Areas using LiDAR Point Clouds." Statistical analyses based on Kolmogorov-Smirnov and Student t-test reveal that the CE-GE method achieves the results for all the evaluation metrics compared to other methodologies. Regarding the confusion matrices of the alternative conceived and the standard categorical cross-entropy method, the smoothed ordinal classification obtains a more consistent classification compared to the nominal approach. Thus, the proposed methodology significantly improves the point-by-point classification of PointNet, reducing the errors in distinguishing between the middle classes (low vegetation and medium vegetation). However, there are some issues should be addressed.
1. In the case of volumetric methods have proposed the partition of a given LiDAR point cloud into regular voxel grids, using 3D Convolutional Neural Networks (3D-CNN) to label each voxel according to the information of its centroid. However, the use of these transformations has been criticized because of the significant computational overhead they introduce, their tendency to increase model complexity, and the potential loss of valuable information. How to solve the problem in this paper?
2. In the field of Forestry Engineering predict the number of strata in three datasets based on ordinal regression techniques to support forest management. However, in the case of airborne LiDAR point clouds, ordinal classification models have not yet been explored. Does this paper propose relevant methods?
3. The second referred to return numbers in order to discriminate ground points from low vegetation points. As previously discussed, the LiDAR point cloud contained six returns. If the return value of a point was higher than or equal to 2, it would mean that the said point could be classified as ground. However, those points with Rn equal to 1 were classified as low vegetation. Will this judgment cause error?
4. The Poisson distribution represents an alternative for modeling the probabilities. However, since the mean and variance of the Poisson distribution are both determined by the parameter λ, it is not possible to center the mean of the distribution in the class interval while achieving a small variance , leading to poor performance. How does this paper solve this problem?
5. Vargas [48] proposed sampling from a beta distribution that is defined within the range of 0 to 1, thus no normal-ization is mandatory and no high variance is achieved. However, an improvement of this distribution could be considered when the number of classes is low. Does this paper use this method? What is the effect? ​​Are there any errors?
Comments on the Quality of English LanguageMinor editing of English language required
Author Response
Response to Reviewers’ comments to manuscript sensors-2905206
The authors gratefully acknowledge the Associate Editor and the Anonymous Reviewers for their detailed and highly constructive criticisms, which greatly helped us to improve the quality and presentation of our manuscript. In the following, we provide detailed, item-by-item, point by point responses to all the very interesting issues raised by the Anonymous Reviewers. We would also like to emphasize that, in order to simplify the review of our manuscript, we have highlighted the main modifications introduced in the reviewed manuscript in blue color to help the Associate Editor and the Anonymous Reviewers find the changes made with regards to the previous version. We are indebted to them for their careful assessment and outstanding suggestions for improving our manuscript, which have been really helpful in order to enhance its presentation and technical quality.
Responses to the comments by Reviewer 3
In this paper, the authors present "Deep Ordinal Classification in Forest Areas using LiDAR Point Clouds." Statistical analyses based on Kolmogorov-Smirnov and Student t-test reveal that the CE-GE method achieves the results for all the evaluation metrics compared to other methodologies. Regarding the confusion matrices of the alternative conceived and the standard categorical cross- entropy method, the smoothed ordinal classification obtains a more consistent classification compared to the nominal approach. Thus, the proposed methodology significantly improves the point-by-point classification of PointNet, reducing the errors in distinguishing between the middle classes (low vegetation and medium vegetation). However, there are some issues should be addressed.
The authors would like to take this opportunity to gratefully thank the Anonymous Reviewer for their precious time and efforts invested in improving this manuscript. The authors have reviewed the English writing of the manuscript and polished its presentation in the reviewed version. Below we provide an item-by-item response to each of these comments and suggestions.
In the case of volumetric methods have proposed the partition of a given LiDAR point cloud into regular voxel grids, using 3D Convolutional Neural Networks (3D-CNN) to label each voxel according to the information of its centroid. However, the use of these transformations has been criticized because of the significant computational overhead they introduce, their tendency to increase model complexity, and the potential loss of valuable information. How to solve the problem in this paper?
The authors gratefully acknowledge for formulating this question. In this study, we employed PointNet for two reasons: (1) based on Qi et al. (2017), this neural network is more computationally efficient than a volumetric 3D CNN; and (2) PointNet operates directly on the 3D point clouds algorithm and does not alter the original nature of the point clouds, thereby reducing potential loss of valuable information. This information can be found on lines 62 to 66 of the reviewed paper.
In the field of Forestry Engineering predict the number of strata in three datasets based on ordinal regression techniques to support forest management. However, in the case of airborne LiDAR point clouds, ordinal classification models have not yet been explored. Does this paper propose relevant methods?
The authors gratefully acknowledge for pointing this out. The present study’s issue differs from Arvidsson and Gullstrand (2021), as the latter have mainly focused on predicting the number of strata (NoS) in forest areas. Unlike this research, they do not classify points into different categories such as ground, low vegetation, medium vegetation, and high vegetation. In this regard, Arvidsson and Gullstrand (2021) explored the stratification (the distribution of tree heights) in forest areas using deep learning approaches. They identified four distinct classes: one-layered forest when all trees have a similar height and a distinct canopy exists (NoS = 1), two-layered forest when two distinct strata with different canopies overlap (NoS = 2), multi-layered forest as an umbrella term for 3 or more strata that are distinct (NoS ≥ 3), and a fully-layered forest when trees of various heights exist and no clear canopy layer can be seen (NoS > 3). Therefore, the methodologies applied in both papers are distinct.
Following the Reviewer’s suggestion, the authors have decided to include the suggested content into the reviewed manuscript.
"This task is different to the classification of points as ground, low vegetation, medium vegetation, and high vegetation, which has not yet been explored." (line 87 to 89).
The second referred to return numbers in order to discriminate ground points from low vegetation points. As previously discussed, the LiDAR point cloud contained six returns. If the return value of a point was higher than or equal to 2, it would mean that the said point could be classified as ground. However, those points with Rn equal to 1 were classified as low vegetation. Will this judgment cause error?
The Reviewer is correct; this filtering could cause errors in areas where the four classes coexist. However, this filtering was specifically used in areas where only two classes coexist: ground and low vegetation. In these areas, the emitted laser pulse of LiDAR technology first encounters the low vegetation and then the ground class. For this reason, this approach assumes that the first return belongs to the low vegetation class and the other returns belong to the ground class. In any case, the authors manually refine the point clouds using CloudCompare.
For clarity, the authors have incorporated the suggested content into the reviewed manuscript.
"This filtering was specifically used in areas where only two classes coexist: ground and low vegetation. In these areas, the emitted laser pulse of LiDAR technology first encounters the low vegetation and then the ground class. For this reason, this approach assumes that the first return belongs to the low vegetation class and the other returns belong to the ground class." (line 154 to 158).
The Poisson distribution represents an alternative for modeling the probabilities. However, since the mean and variance of the Poisson distribution are both determined by the parameter λ, it is not possible to center the mean of the distribution in the class interval while achieving a small variance, leading to poor performance. How does this paper solve this problem?
The authors gratefully acknowledge for pointing this out. This paper introduces the Generalized Exponential Function as a proposed solution to solve the aforementioned problem. In this context, the Poisson distribution was used for comparative analysis. This information can be found on Table 3 (page 11) of the reviewed paper.
Vargas [48] proposed sampling from a beta distribution that is defined within the range of 0 to 1, thus no normalization is mandatory and no high variance is achieved. However, an improvement of this distribution could be considered when the number of classes is low. Does this paper use this method? What is the effect? Are there any errors?
The authors gratefully acknowledge for formulating this question. In this study, the beta distribution proposed by Vargas et al. (2022) was used for comparative analysis (see Table 3 on page 11). This methodology achieved better mean results than all the ordinal methodologies, except the Generalized Exponential Function. In this regard, when the number of classes is small, the Generalized Exponential Function is a better choice, possibly because, according to the analytical methodology proposed in Vargas et al. (2022) to obtain the beta parameters (Figure 2b Vargas et al. (2022)), the values of the beta distribution are more dispersed (flattening or elongation of the distribution) when the number of classes is small (as indicated by the red lines in the figure), which can lead to errors.

Round 2
Reviewer 1 Report
Comments and Suggestions for Authors
The authors have made detailed revisions in response to the comments. I have no further suggestions.
Reviewer 3 Report
Comments and Suggestions for Authors
The authors have solve the problems.
Comments on the Quality of English LanguageMinor editing of English language required